# Myocardial infarction affects Cx43 content of extracellular vesicles secreted by cardiomyocytes

Tania Martins-Marques[1,2,3] , Teresa Ribeiro-Rodrigues[1,2,3] , Saskia C de Jager[4], Monica Zuzarte[1,2,3], Cátia Ferreira[1,3,5] , Pedro Cruz[1,3] , Liliana Reis[3,5], Rui Baptista[1,2,3,5,6], Lino Gonçalves[1,2,3,5] , Joost PG Sluijter[4] , Henrique Girao[1,2,3]

**Ischemic heart disease has been associated with an impairment on intercellular communication mediated by both gap junctions and extracellular vesicles. We have previously shown that connexin 43 (Cx43), the main ventricular gap junction protein, assembles into channels at the extracellular vesicle surface, mediating the release of vesicle content into target cells. Here, using a comprehensive strategy that included cell-based approaches, animal models and human patients, we demonstrate that myocardial ischemia impairs the secretion of Cx43 into circulating, intracardiac and cardiomyocyte-derived vesicles. In addition, we show that ubiquitin signals Cx43 release in basal conditions but appears to be dispensable during ischemia, suggesting an interplay between ischemia-induced Cx43 degradation and secretion. Overall, this study constitutes a step forward for the characterization of the signals and molecular players underlying vesicle protein sorting, with strong implications on long-range intercellular communication, paving the way towards the development of innovative diagnostic and therapeutic strategies for cardiovascular disorders.**

## Introduction

Intercellular communication in the heart can either occur directly, via connexin 43 (Cx43)–containing gap junctions (GJs), or at longer distances, through soluble factors and extracellular vesicles (EVs) (Ribeiro-Rodrigues et al, 2017b; Sluijter et al, 2018; Martins-Marques et al, 2019). Recent studies have shown that Cx43 can also facilitate the communication between EVs and recipient cells, foreseeing relevant and promising therapeutic implications (Soares et al, 2015; Martins-Marques et al, 2016). Whereas GJ ensure the passage of small molecules, including ions, between adjacent cardiomyocytes, contributing for the synchronized heart beating, EV-mediated communication between

cardiac cells is associated with modulation of cardiac development, fibrosis and injury repair, including at the onset of acute myocardial infarction (Sluijter et al, 2018). For example, EVs secreted by cardiomyocyte progenitor cells, mesenchymal stem cells, glucose-deprived H9c2 cardiomyoblasts, and ischemic cardiomyocytes induce endothelial cell proliferation and angiogenesis (Garcia et al, 2015; Vrijsen et al, 2016; Ribeiro-Rodrigues et al, 2017a). Moreover, cardiomyocyte-derived EVs can induce gene expression changes in fibroblasts or activate macrophages (Bang et al, 2014; Paiva et al, 2019).

Despite the impact of EVs on target cells largely depending on their content, the signals whereby this cargo is selectively sorted and dynamically regulated by the pathophysiological state of EV-producing cells remain poorly established (Ribeiro-Rodrigues et al, 2017a; van Niel et al, 2018). Previous studies showed that EV cargo undergoes phosphorylation and modification by ubiquitin and ubiquitin-like proteins, suggesting that posttranslational modifications (PTMs) have an important role in selective sorting of EV proteins (Huebner et al, 2016; van Niel et al, 2018). EV secretion is preceded by the formation of multivesicular bodies (MVBs) that upon fusion with the plasma membrane, release its intraluminal vesicles (ILVs) into the extracellular space. However, these mechanisms are shared with those implicated in the degradative MVB–lysosome pathway, involving recognition of ubiquitinated cargo by the endosomal sorting complexes required for transport (ESCRT) machinery, including tumor suppressor gene 101 (Tsg101) and hepatocyte growth factor–regulated tyrosine kinase substrate (Hrs) (van Niel et al, 2011; Colombo et al, 2013). Nevertheless, what distinguishes secretory from degradative MVBs is currently unknown.

The pathophysiology of acute myocardial infarction and consequent progression into chronic heart failure have been associated with altered trafficking of Cx43-containing GJs, many of these being correlated with changes in the phosphorylation and ubiquitination profile of Cx43 (Fong et al, 2013; Smyth et al, 2014; Martins-Marques et al, 2015a, 2015b, 2015c, 2020; Ribeiro-Rodrigues et al, 2017b). While the impact of ischemia on the modulation of GJ-mediated

---

[1]University of Coimbra, Coimbra Institute for Clinical and Biomedical Research (iCBR), Faculty of Medicine, Coimbra, Portugal   [2]University of Coimbra, Center for Innovative Biomedicine and Biotechnology (CIBB), Coimbra, Portugal   [3]Clinical Academic Centre of Coimbra (CACC), Coimbra, Portugal   [4]Laboratory of Experimental Cardiology, University Medical Center Utrecht Regenerative Medicine Center, Circulatory Health Laboratory, University Medical Center Utrecht, University Utrecht, Utrecht, The Netherlands   [5]Cardiology Department, Centro Hospitalar e Universitário de Coimbra, Coimbra, Portugal   [6]Cardiology Department, Centro Hospitalar Entre Douro e Vouga, Santa Maria da Feira, Portugal

Correspondence: hmgirao@fmed.uc.pt

communication has been a matter of intense research, the consequences of ischemia-triggered events on the levels and/or function of EV-Cx43 channels remain unknown. Therefore, the main aim of this study was to test whether cardiac ischemia affects EV sorting mechanisms and thereby modulate the amount of Cx43 in cardiomyocyte-derived and circulating EVs.

# Results

## Ischemia decreases secretion of Cx43 into circulating EVs in mice subjected to myocardial ischemia/reperfusion (I/R) injury

Although the presence of Cx43 in EVs isolated from cultured cells and peripheral blood has been reported, the impact of ischemia-triggered events on EV-Cx43 sorting remains obscure (Soares et al, 2015; Martins-Marques et al, 2016). Because the molecular signature of circulating EVs may represent an important biomarker of ischemic disease, reflecting alterations within the injured myocardium, we started by isolating serum EVs from a mouse model of I/R injury. In agreement with our previous observations, isolated circulating EVs were positive for the tetraspanins CD63 and CD9, presented typical morphology and EV size and their secretion was up-regulated in animals subjected to I/R (Fig 1A–C) (Deddens et al, 2016; Paiva et al, 2019). Interestingly, the vesicle extract was positive for the specific cardiomyocyte marker Troponin T, suggesting that at least part of these EVs were from cardiac origin (Fig 1A). Results depicted in Fig 1D demonstrate that the levels of Cx43 in circulating vesicles were significantly decreased after 30 min of reperfusion, followed by a recovery at 4 h of reperfusion, while the total amount of EV protein remained constant (Fig S1A).

Because the circulating vesicles profile reflects the contribution of all EVs present in blood, we proceeded to specifically address local generation of EV-Cx43, by isolating interstitial vesicles from the left ventricle of I/R injured mice (Deddens et al, 2016; Loyer et al, 2018). Nanoparticle tracking analysis (NTA) and transmission electron microscopy (TEM) revealed particles with typical EV size and shape (Fig S1B and C). In agreement with the circulating population, Cx43 levels in intracardiac vesicles were significantly reduced after 30 min of reperfusion, recovering at 4 h, whereas the total EV protein levels were not significantly different (Figs 1E and S1D). In both circulating and interstitial EVs, changes in the pan-EV marker GAPDH follow the same pattern of Cx43. Given the limited yield of mouse-derived intracardiac EVs, we confirmed these data in ex vivo Langendorff-perfused rat hearts, subjected to global myocardial ischemia by reduced flow, which better translates the pathological traits of our in vivo model. Heart-derived EVs were positive for Troponin T, CD63, and CD81 and devoid of Calnexin (Fig S1E–G). Analogous to the in vivo model, intracardiac EVs obtained from isolated rat hearts subjected to ex vivo ischemia displayed reduced Cx43 levels, whereas Flotillin-1 levels and EVs amount (NTA) were maintained similar in control and ischemic hearts (Fig S1H).

## Ischemia decreases secretion of EV-Cx43 by cardiomyocytes in vitro

To specifically assess the impact of ischemia on Cx43 levels in cardiomyocyte-derived EVs, we isolated small EVs from HL-1 and

H9c2 conditioned medium by high-speed ultracentrifugation, hereafter referred to as EVs (Martins-Marques et al, 2015b). Our results demonstrate that ischemia impaired the release of Cx43 in EVs derived from both HL-1 and H9c2, after which the amount of EV-Cx43 was gradually restored during reperfusion (Figs 2A and S1I and J). Although the total number of secreted vesicles was not affected by ischemia in HL-1 cells (Fig S1I), the amount of EV-GAPDH and EV-Flotillin-1 was partially decreased, suggesting that besides Cx43, ischemia impairs the release of other EV proteins. Given that endothelial cells are the most abundant cell population in the adult heart, we investigated whether ischemia affects EV-Cx43 secretion by these cells. Interestingly, ischemia decreased Cx43 secretion only in vesicles pelleted by intermediate-speed (16,500$g$), and not by high-speed ultracentrifugation (Fig S2A–D), suggesting that the effect of ischemia upon Cx43 secretion onto EVs is cell-type specific.

Previous studies from our laboratory demonstrated that during ischemia, Cx43 is ubiquitinated and targeted for autophagy degradation after recognition by the adaptor p62 (Bejarano et al, 2012; Martins-Marques et al, 2015c). Hence, to investigate whether decreased Cx43 levels in ischemic EVs were a consequence of Cx43 degradation, we assessed the impact of inhibiting autophagy by silencing p62. Results depicted in Fig 2B show that upon knockdown of p62, the levels of HL1-derived EV-Cx43 in normoxia conditions were increased and partially rescued during ischemia, which suggests that autophagy divert Cx43 from being secreted in EVs, namely, during ischemia. The release of EV-Flotillin-1 was not affected by sip62, nor was the total number of vesicles (Figs 2B and S3A). Nonetheless, the secretion profile of EV-GAPDH was very similar to that of Cx43, suggesting that both proteins share identical mechanisms of EV biogenesis and secretion.

To discard a cell type-specific effect, these results were confirmed in human embryonic kidney (HEK) 293 cells stably over-expressing Cx43 – HEK293^Cx43+ (Fig S3B and C). Given the role of ubiquitin in targeting proteins for autophagy degradation and MVB sorting, these data suggest that during ischemia ubiquitinated Cx43 is preferentially sequestered by p62 which diverts Cx43 to autophagic degradation, rather than sorting into EVs. In accordance, when autophagy was inhibited with knockdown of Atg7 (Xu et al, 2018), the amount of Cx43 secreted in EV was increased (Fig S3D).

## EV-Cx43 is phosphorylated and ubiquitinated

Cx43 PTMs, particularly phosphorylation of Cx43-S373 and Cx43-S368 and ubiquitination, have been associated with the control of GJ-mediated communication, namely, through modulation of protein–protein interactions associated with intracellular trafficking of Cx43 (Smyth et al, 2014; Martins-Marques et al, 2020). Therefore, it is conceivable that phosphorylation and ubiquitination play also a regulatory role on the secretion of EV-Cx43. As indicated in Fig 3A, Cx43 present in EVs was phosphorylated in Thr, Tyr, and Ser residues, namely, S373 and S368 (Fig 3B). Nevertheless, when we overexpressed phospho-null mutants of Cx43, Cx43^S373A, and Cx43^S368A, in parental HEK293^Cx43− cells, EV-Cx43 levels were not affected, suggesting that phosphorylation in these residues is not required for EV-Cx43 secretion (Fig S3E).

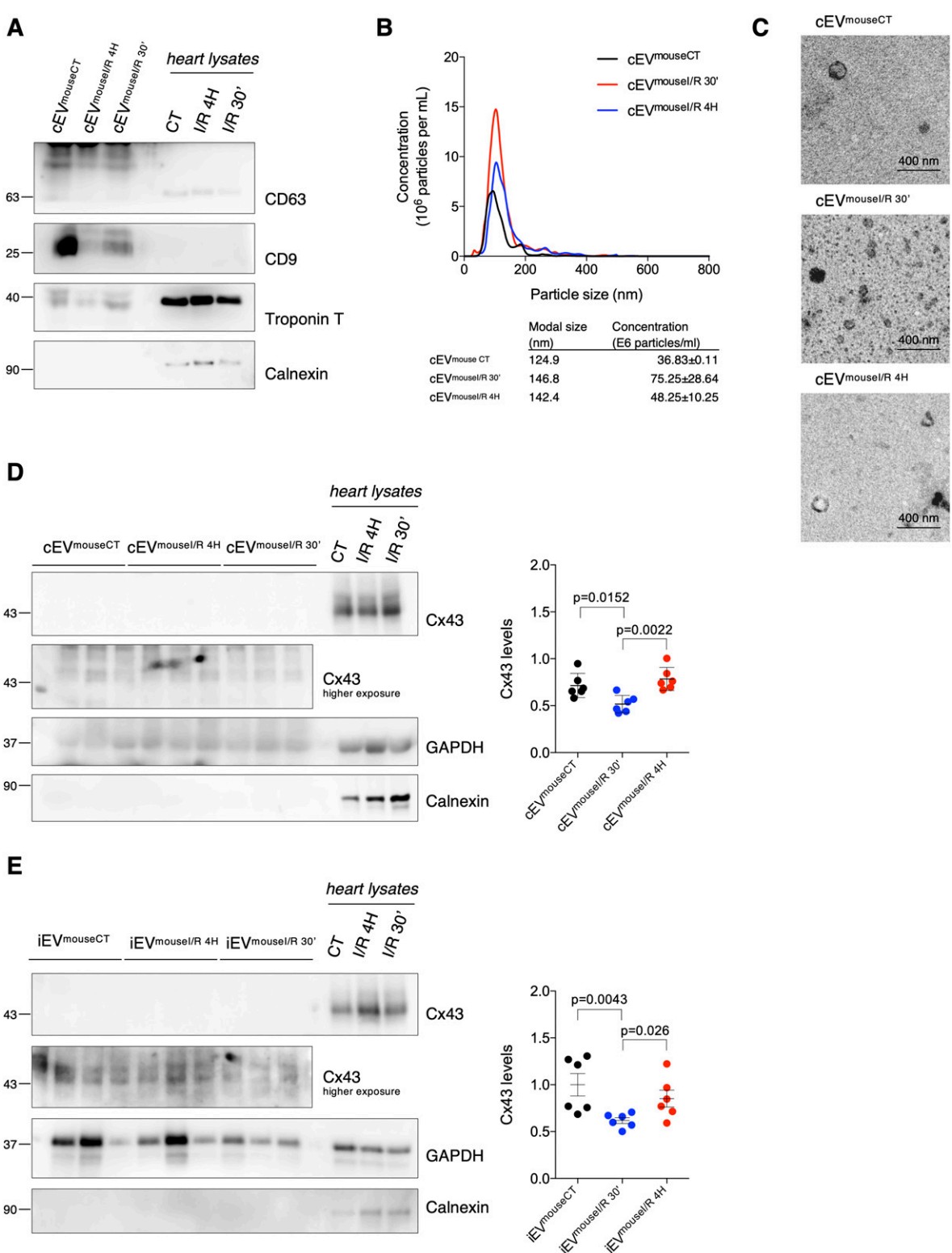

**Figure 1. Ischemia decreases secretion of Cx43 into circulating extracellular vesicles (EVs) in mice subjected to myocardial I/R injury.**
Left coronary artery ligation (60 min) was performed in mice, followed by reperfusion during 30 min (I/R 30') or 4 h (I/R 4H). Sham-operated animals were used as controls (CT). **(A)** Non-reducing WB of circulating EVs (30 μg total protein/lane) from sham (cEV^mouseCT), I/R 4H (cEV^mouseI/R 4H), and I/R 30' (cEV^mouseI/R 30'). CD63 and CD9 were used as positive EV markers, Calnexin as a negative marker and Troponin T as a cardiomyocyte marker. Heart lysates were used as control. **(B)** Nanoparticle tracking analysis of mouse circulating EVs. **(C)** Representative transmission electron microscopy of mouse circulating EVs. **(D)** WB analysis of Cx43 in circulating EVs (30 μg total protein/lane, n = 6). GAPDH was used as a pan-EV marker. **(E)** WB analysis of Cx43 in intracardiac EVs (5 μg total protein/lane) from sham (iEV^mouseCT), I/R 4H (iEV^mouseI/R 4H), and I/R 30' (iEV^mouseI/R 30'; n = 6). Source data are available for this figure.

## A

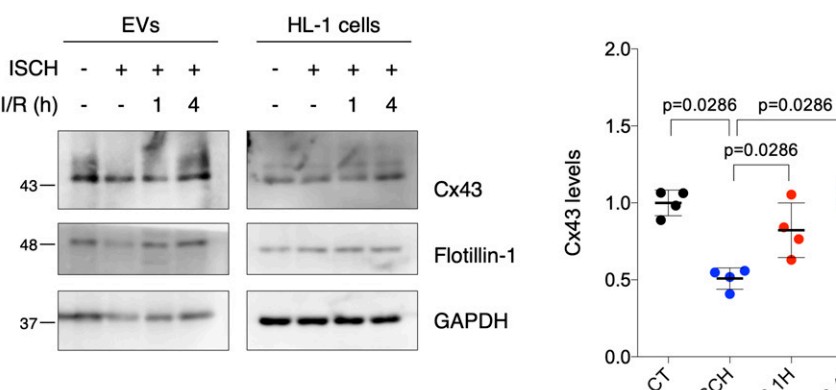

## B

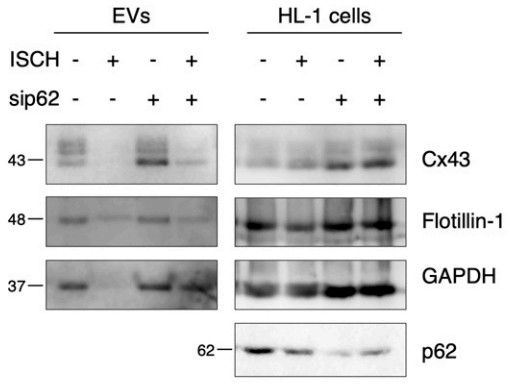

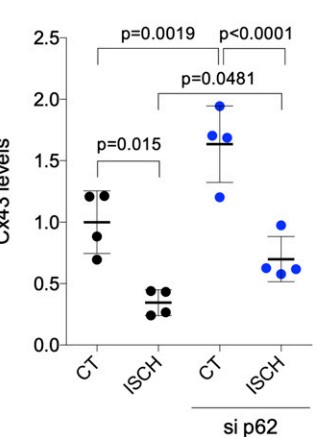

**Figure 2. Ischemia decreases sorting of Cx43 into cardiomyocyte-derived extracellular vesicles (EVs).**
**(A)** WB analysis of EV-Cx43 derived from HL-1 cells subjected to 30-min ischemia (ISCH), followed by 1 or 4 h of reperfusion (I/R). Flotillin-1 and GAPDH were used as pan-EV markers (n = 4). **(B)** WB analysis of EV-Cx43 derived from HL-1 cells subjected to 30 min ischemia and depleted or not of p62 (sip62; n = 4). Source data are available for this figure.

It has been extensively demonstrated that activation of PKC by PMA increases both the phosphorylation and ubiquitination of Cx43, strongly impacting its subcellular distribution (Leithe, 2004). Hence, to investigate its impact upon EV-Cx43 secretion, we exposed HEK293[Cx43+] cells to PMA. Results on Fig 3C and D show that PMA treatment increased the amount of EV-Cx43, which was accompanied by an increase in Cx43 ubiquitination in EV-producing cells. Although in a lesser extent, the total number of secreted vesicles and EV-GAPDH also increased in response to PMA (Fig S3F), whereas EV-Flotillin-1 was maintained (Fig 3C). These results were corroborated in HL-1 cardiomyocytes (Figs 3E and F and S3G). To further confirm the presence of ubiquitinated EV-Cx43, we either immunoprecipitated (IP) Cx43 from EV lysates or pulled-down K63-linked ubiquitinated proteins using tandem ubiquitin binding entities (TUBEs), which revealed that EV-Cx43 was modified by ubiquitin, and more so after exposure to PMA (Figs 3G and S4A) (Ribeiro-Rodrigues et al, 2014).

In accordance with the role of Tsg101 in mediating sorting of ubiquitin-tagged proteins into MVBs, release of EV-Cx43 was impaired after knockdown of Tsg101 (Fig 3H) (Sundquist et al, 2004; Colombo et al, 2013). On the other hand, secretion of Flotillin-1 and GAPDH was not significantly affected after Tsg101 knockdown, whereas the total number of released vesicles was increased (Figs

3H and S4B). Overall, these results suggest that ubiquitin signals the loading of Cx43 into EVs, likely during MVB biogenesis.

To address whether Cx43 found in secreted EVs is of endosomal origin, we overexpressed a constitutively active form of Rab5 (Rab5[Q79L]) that induces the formation of enlarged endosomes containing a large number of ILV-like structures because of endocytic trafficking disruption (Ubelmann et al, 2017). Our results show that although overexpression of GFP-tagged Rab5[Q79L] did not affect basal secretion of EV-Cx43, it significantly impaired the release of Cx43 in response to PMA, when compared with cells expressing GFP alone (Fig S4C). Strikingly, vesicle release was impaired in cells overexpressing Rab5[Q79L] (Fig S4D). Moreover, the percentage of Cx43 within Rab5[Q79L]-positive endosomes was increased in PMA-treated HEK293[Cx43+] and HL-1 cells (Fig S4E and F), suggesting that ubiquitination is promoting the sorting of Cx43 into ILVs that are ultimately released as EVs. Moreover, chemical inhibition of neutral sphingomyelinase down-regulated the release of EV-Cx43 and the total number of vesicles (Fig S4G and H) (van Niel et al, 2018). In addition, when we blocked Cx43 trafficking to the cell surface using Brefeldin A or Cx43 endocytosis with genistein, Cx43 secretion was decreased (Fig S4I and J), strongly suggesting that Cx43 secreted in EVs arises from the plasma

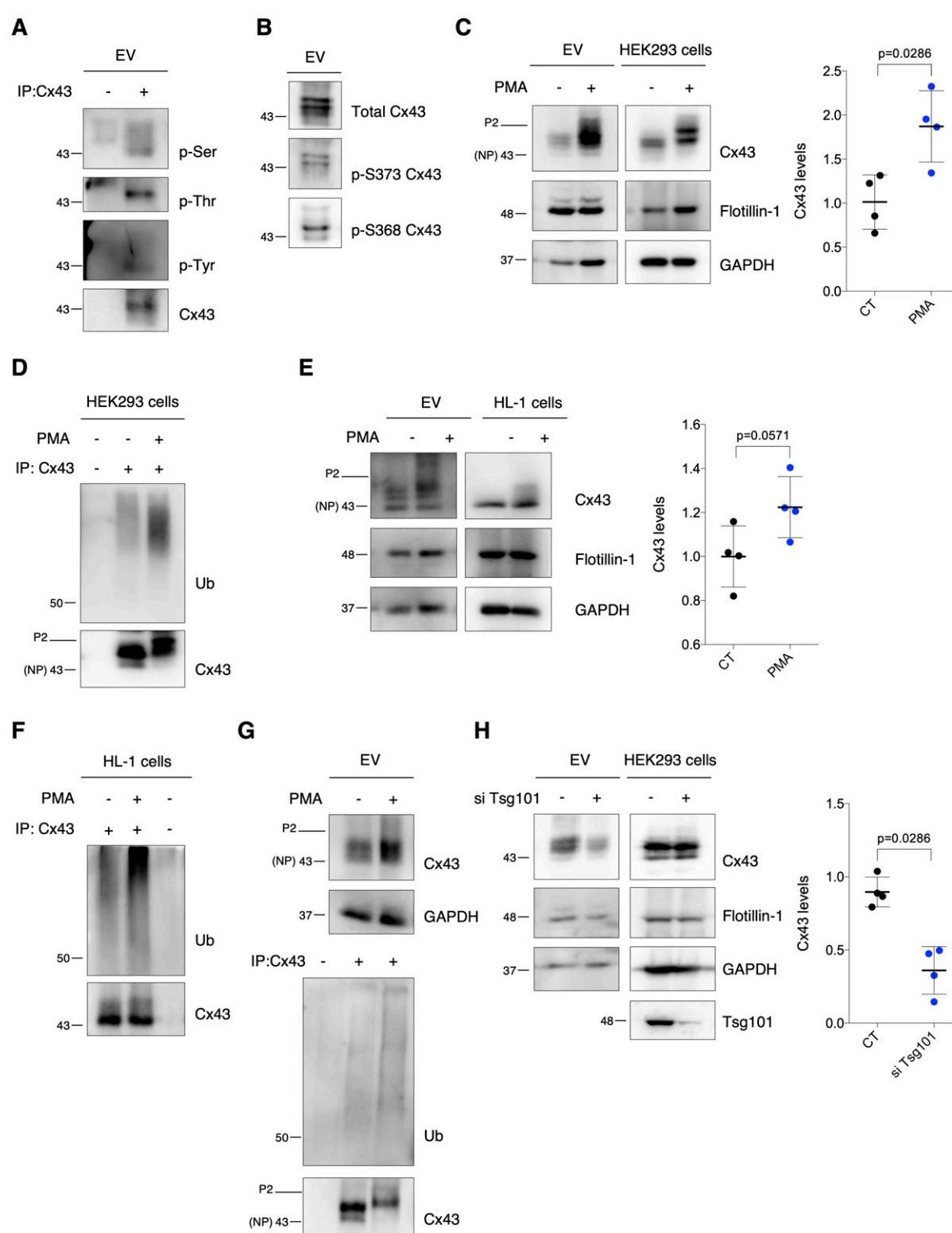

**Figure 3. Extracellular vesicle (EV)-Cx43 is phosphorylated and ubiquitinated.**
**(A)** Cx43 was immunoprecipitated (IP) from HEK293[Cx43+]-derived EVs. Cx43 phosphorylation was evaluated with phospho-Ser, phospho-Thr, and phospho-Tyr antibodies.
**(B)** Phosphorylation of Cx43-S373 and Cx43-S368 was evaluated in HEK293[Cx43+]-derived EVs. **(C)** HEK293[Cx43+] cells were treated with PMA or vehicle for 30 min in EV-depleted medium. EV-Cx43 levels were evaluated by WB (n = 4). **(D)** Cx43 ubiquitination evaluated after IP of Cx43 from HEK293[Cx43+] cells. **(E)** HL-1 cells were treated with PMA or vehicle control for 30 min in EV-depleted medium. EV-Cx43 levels were assessed by WB (n = 4). **(F)** Cx43 ubiquitination evaluated after IP of Cx43 from HL-1 cells.
**(G)** Cx43 ubiquitination evaluated after IP of Cx43 in HEK293[Cx43+]-derived EVs. **(H)** WB analysis of EV-Cx43 derived from HEK293[Cx43+] cells knockdown for Tsg101 (siTsg101), incubated in EV-depleted medium for 8 h (n = 4). NP, non-phosphorylated Cx43; P2, phosphorylated Cx43.
Source data are available for this figure.

membrane, after its internalization and sorting into MVBs (Vercauteren et al, 2010).

## Ubiquitination promotes Cx43 sorting into EVs

To unveil in depth the molecular mechanisms and signals underlying the sorting of EV-Cx43, focusing on the importance of ubiquitin, we inhibited Cx43 ubiquitination in HEK293[Cx43+] cells, either chemically, using the HECT E3 ubiquitin ligase inhibitor, Heclin, or by knockdown of Nedd4, previously shown to catalyze ubiquitin binding to Cx43 (Girão et al, 2009; Mund et al, 2014). Results on Figs 4A and S5A show a decreased amount of EV-Cx43 following inhibition of Cx43 ubiquitination, despite NTA showing an increase in the total number of secreted vesicles (Fig S5B and C). Although in a lesser extent, EV-Flotillin-1 and GAPDH also decreased after siNedd4 (Fig 4A), implicating ubiquitination in regulation of the sorting of other EV proteins. Expectedly, Cx43 ubiquitination was impaired in EV-producing cells incubated either with siNedd4 or Heclin (Figs 4B and S5D). Furthermore, a ubiquitination-defective mutant of Cx43 (Cx43[Y265/286A]; Fig S5E and F) was less present in EVs, whereas Flotillin-1, GAPDH, and the total number of released vesicles remained unaffected (Fig S5G and H) (Catarino et al, 2011; Fong et al, 2013; Ribeiro-Rodrigues et al, 2017b).

Besides ubiquitin attachment, we also modulated Cx43 ubiquitination at the level of deubiquitinating enzymes (DUBs), namely, DUB-associated molecule with the SH3 domain of STAM (AMSH) and ubiquitin-specific peptidase 8 (USP8), known to catalyze the removal of ubiquitin moieties from Cx43 (Ribeiro-Rodrigues et al, 2014; Sun et al, 2018). Our results show that siRNA-mediated knockdown of USP8 increased EV-Cx43 release, as well as the total number of vesicles, whereas depletion of AMSH decreased EV-Cx43 levels (Figs 4C and S5I). Secretion of GAPDH, but not EV-Flotillin-1 follows the same profile as EV-Cx43.

Taking into account the results obtained up to this point, it would be plausible to suggest that decreased Cx43 levels in ischemic EVs result from an impaired Cx43 ubiquitination. Nonetheless, our data demonstrate that ubiquitination of intracellular Cx43 was not affected in ischemic EV-producing cells (Fig S5J). Moreover, the release of EV-Cx43 in ischemia was not significantly disturbed by pre-treatment with the ubiquitination inhibitor Heclin, suggesting that ischemia-induced ubiquitination of Cx43 results in early commitment with degradation, even before a significant Cx43 decrease, ultimately restraining the release of EV-Cx43 (Fig S5K).

## I/R impacts retention of EV-Cx43 in the ECM

Ample evidence demonstrates that EV–ECM associations facilitate EV binding and uptake by target cells and may limit long-range vesicle diffusion, acting as reservoir to enable EV release after injury-induced ECM remodeling, including during ischemia and I/R (Buzás et al, 2018). To address whether EV-Cx43 can be retained in the ECM, we performed decellularization of post-confluent HEK293[Cx43+] cultures, followed by collagenase digestion and EV isolation by differential ultracentrifugation (Chaudhary et al, 2016). TEM revealed

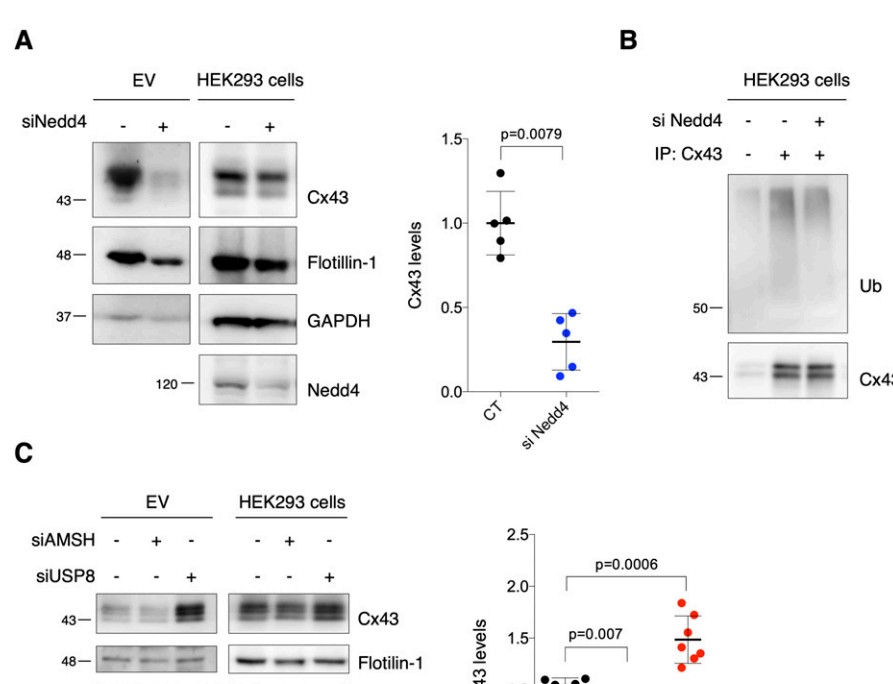

**Figure 4. Ubiquitination promotes secretion of extracellular vesicle (EV)-Cx43.**
**(A)** WB analysis of EV-Cx43 derived from HEK293[Cx43+] cells knockdown for Nedd4 (siNedd4), incubated in EV-depleted medium for 8 h (n = 5). **(B)** Cx43 ubiquitination in HEK293[Cx43+] cells evaluated after immunoprecipitation of Cx43. **(C)** WB analysis of EV-Cx43 derived from HEK293[Cx43+] cells knockdown for AMSH (si AMSH) or USP8 (siUSP8), incubated in EV-depleted medium for 8 h (n = 7).
Source data are available for this figure.

vesicles with typical size and morphology, with preserved binding to ECM fibers (Fig S6A). However, lower molecular weight bands (<30 kD) were observed in the Western Blot (WB) for Cx43, likely reflecting Cx43 cleavage by collagenase (Fig S6B and C). To overcome this issue, we proceeded to evaluate the presence of EVs in total ECM extracts, without collagenase digestion and ultracentrifugation, which rendered a heterogeneous population of vesicles and ECM proteins, but prevented enzymatic cleavage of Cx43 (Fig S6B, lane 2). ECM extracts were positive for canonical EV markers, devoid of the nuclear marker p53 and presented intact vesicles and ECM fibers (Fig 5A and B), reinforcing the suitability of this method to evaluate Cx43 levels in ECM-bound vesicles.

Next, we sought to assess the impact of ischemia and ubiquitination on the amount of Cx43 in HEK293$^{Cx43+}$-derived ECM. Similar to the observed in conditioned medium-derived EVs, Cx43 levels in the ECM decreased by both Heclin treatment and ischemia (Fig S6D and E). To address whether the reduction of ECM-associated Cx43 reflected an inhibited Cx43 secretion or a decreased binding of Cx43-vesicles to ECM fibers, we evaluated EV/collagen binding in vitro. Our results show that the total amount of vesicles (Flotillin-1 and GAPDH-positive) bound to collagen fibers was similar in all experimental conditions, whereas differences in Cx43 mirrored its levels in input EVs (Fig S6F and G). To assess whether surface EV-Cx43 affects binding to the fibers, we performed this assay using vesicles with or without Cx43. The lack of Cx43 did not interfere with EV-collagen binding, corroborating previous results (Fig S6H).

Next, we asked whether I/R injury impacts on the amount of EV-Cx43 retained in ECM derived from HL-1 cardiomyocytes (Fig S7A). In agreement with data presented above, ECM-associated Cx43 levels decreased after ischemia, being gradually recovered during reperfusion (Fig 5C), which was confirmed in mixed primary cultures of cardiomyocytes and fibroblasts (Fig S7B–D). In addition, we performed decellularization of hearts from I/R injury mice (Silva et al, 2016). Scanning electron microscopy revealed that ECM fibers were well-preserved, whereas TEM showed the presence of intact vesicles (Fig S7E). Masson's trichrome evidenced the blue network of collagen fibers, reinforcing the preservation of ECM structure following our protocol (Fig S7E). Moreover, CD63 was detected in myocardium-derived ECM, but not Calnexin, suggesting that we co-isolated EVs with limited cellular contamination (Fig 5D). Surprisingly, our results show that ECM-associated Cx43 levels increased in mice subjected to 30 min of reperfusion, being reduced after 4 h, whereas the total amount of ECM protein remains unaltered (Figs 5E and S7F).

### Cx43 decreases in circulating EVs from patients with ST-segment elevation myocardial infarction (STEMI)

To extend the relevance of our findings to a human pathophysiological context, we evaluated Cx43 levels in circulating EVs isolated from human serum of STEMI patients (hEV$^{STEMI}$) or control subjects (hEV$^{CT}$; Table 1). As with mouse samples, polymer-based EV isolation successfully precipitated most CD63 and CD81-EVs from serum samples, devoid of Calnexin and positive for the cardiomyocyte marker Troponin T, with a canonical cup-shaped morphology (Figs S8A and B and 6A and B). When comparing EVs isolated from

controls with STEMI patients, EV-protein content appeared similar between the two populations, whereas circulating hEV$^{STEMI}$ presented higher concentration and modal size (Fig S8C). Next, we quantified the amount of Cx43 in circulating EVs by ELISA. Results on Fig 6C show that Cx43 levels in EVs from STEMI patients' serum were lower when compared with the control population. A similar pattern was observed when plasma samples were used to assess the levels of circulating EV-Cx43 (Fig S8D–F). This decrease in Cx43 levels in hEV$^{STEMI}$ was specific, as other EV proteins, such as Alix, Hsp90, and GAPDH remained constant (Fig 6D). To further validate our findings, we performed size-exclusion chromatography (SEC) and differential ultracentrifugation to isolate circulating EVs, which further confirmed that the levels of Cx43 in circulating EV decreased in STEMI patients (Fig S8E and F).

Finally, we sought to assess the functionality of Cx43 channels in EVs isolated from serum of human controls, using a modified sucrose-density gradient ultracentrifugation assay, in which EVs containing permeable Cx43 channels migrate to lower positions in the tube (higher sucrose concentration), whereas impermeable EV-Cx43 remain in upper fractions (lower sucrose concentration) (Bao et al, 2004; Fiori et al, 2014). Our results show that circulating EV-Cx43 were mostly found in upper fractions (i.e., were predominantly closed; Fig 6E). The presence of Hsc70 in the same fractions as Cx43 suggests that this technique is suitable to assess EV-Cx43 channel permeability. Because Cx43 phosphorylation is a major regulator of GJ channel permeability, with dephosphorylation of Cx43-S368 being reported as an inducer of Cx43 channel opening in vitro and in vivo, we validated this approach in HEK293 cells overexpressing either *wild type*, phosphorylation-null mutants of Cx43, or a closed channel mutant (Cx43$^{T154A}$) as control (Beahm et al, 2006; Fiori et al, 2014; Ribeiro-Rodrigues et al, 2017b). Our data demonstrate that EV-Cx43$^{WT}$ and EV-Cx43$^{S373A}$ channels were mostly closed, resembling the profile of EV-Cx43$^{T154A}$, whereas EV-Cx43$^{S368A}$ migrated towards higher sucrose densities, consistent with a higher opening probability for these channels (Fig S9A). As previously reported for Cx43 hemichannels, EV-Cx43 channel closure could be induced by low pH levels (Fig S9B) (Fiori et al, 2014).

## Discussion

In recent years, orchestrated cell–cell communication via EVs has gained increased interest, as vesicles can carry a specific repertoire of information between different cells and tissues. Nevertheless, the regulatory mechanisms underlying the selective sorting of EV contents remain poorly established. Because the content of released EVs can mirror the pathophysiological state of cells, particular clinical relevance has been attributed to circulating EVs, which may serve as potential disease biomarkers. Therefore, it is crucial to elucidate the mechanisms regulating the sorting of biomolecules into EVs. In the present study, using cell-based, in vivo and ex vivo animal models and human patients, we demonstrate that ischemia impairs the sorting of Cx43 into cardiomyocyte-derived EVs and circulating vesicles. Although additional studies are required, these results may open new avenues

**Figure 5. I/R impacts retention of extracellular vesicle (EV)-Cx43 in the ECM.**
**(A)** Non-reducing WB of HEK293$^{Cx43+}$-derived ECM (10 μg total protein/lane). CD63, CD81, CD9, Tsg101, Flotillin-1, and GAPDH was used as EV markers. p53 was used to determine cellular contamination. **(B)** Representative transmission electron microscopy of HEK293$^{Cx43+}$-derived ECM. **(C)** Levels of Cx43 in ECM (10 μg total protein/lane) from HL-1 cells subjected to 30 min ischemia (ISCH), 1 h or 4 h of reperfusion (I/R; n = 4). **(D)** Non-reducing WB of heart-derived ECM (10 μg total protein/lane) from sham (ECM$^{mouseCT}$), I/R 4H (ECM$^{mouseI/R\ 4H}$), and I/R 30' (ECM$^{mouseI/R\ 30'}$). **(E)** WB analysis of Cx43 in heart-derived ECM (10 μg total protein/lane; n = 6).
Source data are available for this figure.

**Table 1. Demographic and clinical data from human subjects.**

| | Control (n = 29) | STEMI (n = 28) | *P*-value |
|---|---|---|---|
| Age, years | 64 (53–67) | 65 (57–73) | 0.506 |
| Female | 11 (37.9) | 5 (17.9) | 0.092 |
| Ethnicity—White | 29 (100) | 28 (100) | >0.999 |
| Height, cm | 167.8 ± 8.5 | 168.0 ± 6.7 | 0.947 |
| Weight, kg | 80.0 (73.0–87.5) | 77.5 (70.0–88.5) | 0.856 |
| BMI, kg/m$^{-2}$ | 28.4 (27.2–30.1) | 28.3 (25.6–29.6) | 0.854 |
| CV risk | | | |
| Type 2 diabetes *mellitus* | 7 (24.1) | 6 (21.4) | 0.808 |
| Arterial hypertension | 24 (82.8) | 15 (53.6) | 0.018[a] |
| Dyslipidemia | 24 (82.8) | 18 (64.3) | 0.113 |
| Active smoker | 10 (34.5) | 15 (53.6) | 0.147 |
| Previous CAD | 0 (0) | 5 (17.9) | 0.017[a] |
| Chronic kidney disease | 0 (0) | 5 (17.9) | 0.017[a] |
| CCU admission | | | |
| Troponin I, ng/ml | N/a | 25.8 (4.2–54.9) | — |
| Peak troponin I, ng/ml | N/a | 65.0 (24–117.0) | — |
| NT-proBNP, pg/ml | N/a | 488.5 (180.8–1,642.0) | — |
| C-reactive protein, mg/dl | N/a | 1.1 (0.3–2.9) | — |
| Serum creatinine, mg/dl | N/a | 0.9 (0.8–1.3) | — |
| Hemoglobin, g/dl | N/a | 13.44 ± 2.03 | — |

Values are median (interquartile range), n (%) or mean ± SD; BMI, body mass index; CV, cardiovascular; CAD, coronary artery disease; CCU, coronary care unit; NT-proBNP, N-terminal prohormone of brain natriuretic peptide; N/a, non-applicable.
[a]Adjusted *P*-value < 0.05.

for establishing Cx43 levels in circulating EVs as a systemic marker of cardiac injury. Furthermore, we identified ubiquitin as a signal that targets Cx43 into EVs. Given the role played by ubiquitin in delivering proteins to the MVB, an intermediate organelle shared between exosome secretion and lysosomal degradation, it would be important to unveil how these different pathways determine the final fate of Cx43. Accordingly, we show that the autophagy adaptor p62 can act as a putative regulator of the balance between Cx43 degradation and secretion.

Our data show that Cx43 levels were significantly decreased in circulating EVs 30 min after the start of reperfusion in ischemic animal models, with a partial recovery at longer reperfusion periods. Assuming that only a small percentage of circulating EVs is of cardiac origin, it is conceivable that time-dependent systemic changes, following ischemia, affect the release of EV-Cx43 by different cell types, besides cardiomyocytes. We also observed a down-regulation of circulating EV-Cx43 in STEMI patients. However, we acknowledge the limitations of our study that included a small human patient cohort, in which the ischemia time window was not as well-defined as in the animal model. Moreover, it is plausible that the causative factors for decreased EV-Cx43 in I/R are different depending on the time window after injury. For instance, lower levels of circulating EV-Cx43 in early reperfusion periods might constitute a strategy to maintain cellular Cx43 levels, to preserve intercellular communication, whereas following longer periods, a decrease of Cx43 in circulating EVs mirrors a general degradation of Cx43 in EV-secreting cells.

Although the consequences of having less Cx43 in ischemic EVs were not assessed in the present work, this may ultimately affect long-distance cell–cell communication. Grounded on our previous studies showing that EV-Cx43 facilitates the communication with acceptor cells and because post-infarction circulating EVs exacerbate inflammatory responses, contributing to impaired heart function, we can speculate that reduced Cx43 levels prevent the spreading of certain metabolites or signaling molecules, constituting a cardioprotective mechanism (Soares et al, 2015; Biemmi et al, 2020).

The presence of Troponin T in circulating EVs is consistent with a cardiomyocyte origin. However, circulating vesicles may derive from a multitude of different cell types and organ locations, and even within the heart, EVs can arise either from ischemic or non-ischemic areas (Sluijter et al, 2018; Théry et al, 2018). Nonetheless, our in vitro data showing that EVs secreted by ischemic cardiomyocytes, but not endothelial cells, present lower Cx43 levels, advocates in favor of a cardiomyocyte-specific response to ischemia.

In the present study, we also provide evidence that Cx43 channels in circulating EVs were mostly closed. According to our model, EV-Cx43 channels will only open after docking with Cx43 hemichannels at the plasma membrane of acceptor cells, to release intraluminal cargo in a controlled manner and prevent leakage. Notwithstanding this, ischemia-induced alterations in the cellular microenvironment, including low pH levels, may alter Cx43 channel gating during EV biogenesis and/or docking with recipient cells, thereby impacting EV-cell communication.

**Figure 6. Cx43 levels decrease in circulating extracellular vesicles (EVs) from STEMI patients.**
**(A)** Representative transmission electron microscopy of circulating human EVs from control (hEV[CT]) and STEMI patients (hEV[STEMI]). **(B)** Representative WB of circulating EVs (30 μg total protein/lane). Heart lysates were used as control. **(C)** Levels of Cx43 were evaluated in hEV[CT] and hEV[STEMI]. Individual levels, median, and interquartile range are plotted on graph (n = CT, n = 28 STEMI). **(D)** WB analysis of Cx43, Alix, Hsp90, and GAPDH in circulating EVs (30 μg total protein/lane). **(E)** Permeability of EV-Cx43 channels in circulating vesicles from human controls, assessed by sucrose-based transport-specific density shift.
Source data are available for this figure.

It is recognized that the ECM represents an important reservoir of EVs, which can be released after injury-induced ECM disruption (Buzás et al, 2018). Despite its importance, little attention has been given to this issue, with most of the studies focusing on the vesicles secreted into the extracellular milieu, either in cell culture medium or biological fluids, neglecting the impact of EVs retained within the ECM. Strikingly, our data demonstrate that the levels of ECM-associated Cx43 were reduced after in vitro ischemia, a phenotype that might merely reflect decreased secretion of EV-Cx43 by producing cells. Surprisingly, in our I/R injury mice model, ECM-associated Cx43 levels increased after 30 min of reperfusion, which may be caused by changes in the biophysical/biomechanical properties of ECM, preventing long-range diffusion of EV-Cx43 (Rackov et al, 2018; Lenzini et al, 2020). Because the profile of both Cx43 and GAPDH in ECM extracts was similar in these conditions, it is also possible that modifications on EV protein surface content might favor in-bulk vesicle attachment to the ECM. Contrarily, Cx43 levels decreased in intracardiac EVs that likely represent the interstitial vesicles population, loosely attached to the ECM. Despite our results showing that Cx43 did not affect EV-collagen interaction in vitro, we can speculate that Cx43 modulates

EV–ECM binding in specific conditions, including within the ischemic environment. The discrepancies among our in vivo, ex vivo, and in vitro data can be explained by various factors. For example, although our results demonstrate that ECM extracts were positive for CD63 and CD81 and devoid of intracellular contaminants, we cannot completely ensure that these vesicles do not partially result from artifactual release of intracellular contents during tissue processing, particularly during decellularization (Théry et al, 2018; Ilahibaks et al, 2019).

A recent proteomic study found that ~13% of the proteins identified in EVs derived from urine are ubiquitinated (Huebner et al, 2016). Moreover, C-terminally ubiquitin fusion can target EV secretion of soluble proteins (Cheng & Schorey, 2016). In accordance, our results demonstrate that EV-Cx43 was ubiquitinated, ascribing to this modification an important role in Cx43 secretion. Importantly, as our data suggest that EV-Cx43 is of plasma membrane origin, ubiquitin signaling may be required not only for internalization, but also for post-endocytic sorting of Cx43 and incorporation at the MVB interface, through interaction with ESCRTs, including Tsg101. Nonetheless, previous data from our group demonstrated that ischemia-induced ubiquitination targets degradation of Cx43-containing

GJ (Martins-Marques et al, 2015c). In this work, we show that although ischemia maintained the levels of ubiquitinated Cx43 in EV-producing cells, its secretion was significantly decreased, suggesting that ubiquitin-dependent Cx43 secretion is modulated by the pathophysiological context. Overall, we demonstrate that ischemia impaired the secretion of EV-Cx43, likely by targeting ubiquitinated Cx43 for degradation, rather than EV release. Impairment of degradation pathways may diverge Cx43 from degradation into EV sorting, as our data following the depletion of either p62 or Atg7 suggest. ATP and/or oxygen depletion during ischemia may also contribute to altered activation of signaling pathways, resulting in impaired secretion of Cx43 that is gradually resumed during reperfusion.

Given that ubiquitinated proteins are found in EVs, this implies that EV cargo can evade deubiquitination or involves ESCRT-independent biogenesis mechanisms. Previous studies from our laboratory have shown that AMSH-mediated deubiquitination of Cx43 protects GJ from degradation (Ribeiro-Rodrigues et al, 2014). Here, depletion of AMSH decreased EV-Cx43 levels, suggesting that lysosomal degradation of Cx43 was promoted, whereas knockdown of USP8 up-regulated the release of ubiquitinated Cx43 in EVs. Thus, we propose that, in the absence of AMSH, ubiquitinated Cx43 is mainly directed to degradation, while lack of USP8 results in the accumulation of Cx43, likely at late endosomes/MVBs, modified with ubiquitin chains that favor its sorting into ILVs that will be subsequently secreted as EVs (Bejarano et al, 2012; Martins-Marques et al, 2015c).

Despite the use of NTA throughout this article as a measure of total vesicle number, we recognize the limitations of this approach that may overestimate vesicle quantification because of detection of non-EV particles, namely protein aggregates. However, currently available methods to assess the amount of EVs are poorly reliable when used without complementary assays, including the gold standard measure of total protein content or the levels of specific EV markers (Théry et al, 2018). It is also important to recognize that in some circumstances, as observed in circulating and intracardiac vesicles obtained from I/R-injured mice, total protein content do not correlate with EV numbers, likely reflecting stimuli-induced selective sorting mechanisms. Hence, there is not a trustworthy EV protein marker that can be used as loading control for the vesicle fraction. In agreement, although in the vast majority of our experiments, the levels of EV-Flotillin-1 were not affected by the different treatments, secretion of EV-GAPDH followed a profile identical to that of EV-Cx43, suggesting that part of the EV biogenesis and secretion mechanisms are shared by both proteins.

The findings gathered in the present study demonstrate that Cx43 secretion into cardiomyocyte-derived EVs is impaired during myocardial I/R injury, which reflects in the amount of Cx43 in circulating vesicles from animal models and STEMI patients. Altogether, our results suggest that a fine balance between PTMs, protein adaptors and the physiological state of vesicle-producing cells are of utmost importance for the regulation of Cx43 sorting into EVs, which may open new avenues towards the search for novel disease biomarkers.

# Materials and Methods

## Cell cultures

H9c2 (Sigma-Aldrich) and HEK-293 cells were cultured in DMEM (Gibco, Thermo Fisher Scientific), with 10% FBS (Gibco, Thermo Fisher Scientific)

and 1% Penicillin/Streptomycin (Pen/Strep). HEK293$^{Cx43+}$ cell line was established as previously described (Catarino et al, 2011). HL-1 cells (clone 6) were obtained from Dr Emmanuel Dupont (Imperial College London), established from the HL-1 parental cell line (Dias et al, 2014). HL-1 cells were cultured in gelatin/fibronectin (0.02% gelatin/0.1% fibronectin) coated culture vessels and maintained in Claycomb medium (Sigma-Aldrich) with 0.1 mM Norepinephrine, 2 mM L-Glutamine, 10% FBS and 1% Pen/Strep. Mouse Cardiac Endothelial Cells were obtained from Dr Justin C Mason (Imperial College London). Mouse Cardiac Endothelial cells were maintained in DMEM with 10 mM Hepes, 5% FBS, and 1% Pen/Strep. All cells were maintained at 37°C under 5% CO$_2$.

## Cell treatments

Ischemia was simulated by incubation in a ischemia-mimetic solution (118 mM NaCl, 4.7 mM KCl, 1.2 mM KH$_2$PO$_4$, 1.2 mM MgSO$_4$, 1.2 mM CaCl$_2$, 25 mM NaHCO$_3$, 5 mM lactate, 20 mM 2-deoxy-D-glucose, and 20 mM Na-HEPES, pH 6.6) in hypoxic pouches (GasPakTM EZ; BD Biosciences), equilibrated with 95%N$_2$/5%CO$_2$ (Martins-Marques et al, 2015c). In reperfusion experiments, ischemic buffer was replaced by complete medium, and cells returned to normoxia. 100 µM Heclin (Tocris), 5 µg/µl PMA (tumor-promoting phorbol ester 12-O-tetradecanoylphorbol-13-acetate), 50 µM genistein, or 4 µM GW4869 (Sigma-Aldrich) were used.

## Cell transfection

Transfections were performed with Lipofectamine 2000 (Thermo Fisher Scientific) according to the manufacturer. Briefly, siRNA (20 nM final concentration) transfections were performed twice at intervals of 24 h. Experiments were performed 48 h after the first transfection, with exception of siAMSH and siUSP8, performed 72 h after transfection. Non-targeting sequences (Ambion, Thermo Fisher Scientific) were used as controls. siRNA against Nedd4 (s9416, GGAAGAUCCAAGAUUGAAAtt; s9417, GGCGAUUUGUAAACCGAAUtt), AMSH (s20852, CAUCCUCUAUAACCAAGUAUtt; s20853, GAGUUGAGAUUAUCCGAAUtt), Tsg101 (s14440, CUGUCAAUGUUA-UUACUCUtt; s14441, GAGACCUAACUGUACGUGAUtt), and p62 (s71143, CCA-AUGUCAAUUUCCUGAAtt; s7144, GGAACUCGCUAUAAGUGCAtt) were from Ambion (Silencer Select Pre-designed siRNA). siRNA against USP8 were obtained from Dharmacon Inc (Lafayette; D-005203-02, UGAAAUACGU-GACUGUUUAUU; D-005203-03, GGACAGGACAGUAGAUAUU). For Atg7 knockdown, HL-1 cells were transduced with an empty lentiviral vector or a lentiviral vector containing shRNA targeting mouse Atg7 for 7 d (Ferreira et al, 2013). Both lentiviral vectors were provided by Dr. A.M. Cuervo (Albert Einstein College of Medicine, Yeshiva University). Plasmids encoding for GRP-Rab5$^{Q79L}$ were kindly provided by Dr Cláudia G Almeida (Chronic Diseases Research Center [CEDOC], NOVA Medical School, NOVA University Lisbon) (Ubelmann et al, 2017). Cx43 mutants were generated by site-directed mutagenesis using the pENTR (Cx43$^{Y265/286A}$ and Cx43$^{T154A}$) or the pENTR-V5-Cx43 plasmid (V5-Cx43$^{S368A}$ and V5-Cx43$^{S373A}$) and verified by sequencing (Soares et al, 2015).

## Animal models

### Ex-vivo Langendorff heart perfusion model

10-wk-old Wistar rats, obtained from our local breeding colony, were handled according to European Union guidelines (2010/63/

EU), approved by ORBEA-IBILI (permit 13/2015). Animals were anaesthetized with 85 mg/kg ketamine and 10 mg/kg xylazine and heparinized. Hearts were perfused ex vivo (perfusion pressure of 70 mm Hg, constant flow rate of 11 ml/min/g wet weight), with modified Krebs–Henseleit buffer (118 mM NaCl, 25 mM NaHCO$_3$, 4.7 mM KCl, 1.2 mM MgSO$_4$, 1.2 mM KH$_2$PO$_4$, 10 mM Hepes, 1.25 mM CaCl$_2$, and 10 mM glucose, pH 7.49), equilibrated with 95%O$_2$/5%CO$_2$ at 37°C. Perfusion was stabilized for 10 min, after which hearts were perfused for further 30 min (control) or subjected to global 30 min of low flow ischemia, by reducing the perfusion flow to 20% of the pre-ischemia value. Hearts were snap-frozen in liquid N$_2$ and stored at –80°C.

### *Left coronary artery ligation mice model*
Animal experiments were approved by the Ethical Committee on Animal Experimentation of the University Medical Center Utrecht, conform to the "Guide for the care and use of laboratory animals." Healthy female Balb/c mice (10–12 wk, 18–21 g) received standard chow and water ad libitum. Mice were anaesthetized by i.p. injection of medetomidinehydrochloride (1.0 g/kg), midazolam (10.0 mg/kg) and fentanyl (0.1 mg/kg), intubated and connected to a respirator (1:1 oxygen-air ratio, times/minute). A core body temperature of 37°C was maintained by rectal monitoring and an automatic heating blanket during all procedures. The heart was accessed through left lateral thoracotomy with pericardium incision. The left coronary artery was ligated for 60 min with 8-0 Ethilon suture (Ethicon) and polyethylene-10 tubing section placed over the left coronary artery to allow vessel reopening. Surgical wounds were closed, followed by i.p. injection of antagonist (atipamezole hydrochloride [3.3 mg/kg], flumazenil [0.5 mg/kg]), and analgesia (buprenorphin [0.15 mg/kg]). Animals were euthanized 30 min and 4 h after reperfusion. Blood was collected through orbital bleeding. To prevent loss of myocardial EVs, no in situ perfusion was performed. Myocardium was excised, left ventricle was isolated, and separated into two halves of which were snap frozen (for intra-cardiac EV isolation) or stored in optimal cutting temperature (TissueTek, Sakura Finetek, Alphen aan de Rijn; for decellularization) and kept at –80°C until further processing.

### Isolation of primary cultures of cardiomyocytes and fibroblasts

Primary cultures of cardiomyocytes and fibroblasts were isolated from Wistar rats, obtained from our local breeding colony. Briefly, hearts excised from neonatal rats (P3-P5) were subjected to 0.1% trypsin–EDTA digestion overnight, at 4°C. Type II collagenase (75 U/ml; Gibco, Thermo Fisher Scientific) digestion was further performed for 30 min, at 37°C, followed by mechanical dissociation of the tissue and enzyme inactivation by the addition of DMEM containing 10% FBS. Digested tissues were transferred through a screen (70 $\mu$m), and cells were recovered by centrifugation and plated into 1% (wt/vol) gelatin-coated dishes for 3 h. After that, non-adherent cells (enriched in cardiomyocytes) were plated in fibronectin-coated dishes and maintained in DMEM, supplemented with 10% FBS, 1% penicillin/streptomycin (100 U/ml:100 $\mu$g/ml), at 37°C, under 5% CO$_2$. Cells were maintained in culture for 10 d to allow fibroblast proliferation and matrix deposition before experiments were performed.

### Human samples

Patients were included if they had a clinical diagnosis of ST-segment elevation myocardial infarction (STEMI), following the Fourth Universal Definition of Myocardial Infarction Guidelines (Thygesen et al, 2019). Exclusion criteria included age below 18 or above 85 yr, pregnancy, shock or acute pulmonary oedema before admission, active oncological disease, end-stage kidney disease, severe cerebrovascular disease, severe anaemia or coagulopathy. Peripheral blood samples were drawn from (i) STEMI patients during the first 12 h before reperfusion by primary percutaneous coronary intervention and (ii) from age- and sex-matched control patients with no epicardial coronary artery disease. The protocol followed the Declaration of Helsinki (2008), with the approval from Coimbra Hospital and University Center research ethical committee (#CHUC-057-15). All patients filled a written informed consent form (Table 1).

### EV isolation

Cells were cultured in EV-depleted medium, prepared by ultracentrifugation of 50% FBS (120,000*g*, 16 h) (Théry et al, 2006; Soares et al, 2015). Conditioned medium was subjected to differential centrifugation at 4°C (10 min, 300*g*; 20 min, 16,500*g*). Supernatants were filtered (0.22 $\mu$m) and ultracentrifuged (70 min, 120,000*g*). Unless stated otherwise, WB analysis were performed in EVs obtained from 12 × 10$^6$ secreting cells, during the indicated conditioning times and treatments. Intra-cardiac EVs were isolated after tissue mincing in 0.9% NaCl and centrifugation (5 min, 400*g*; 15 min, 2,000*g*; and 45 min, 16,500*g*). Supernatants were incubated with Total Exosome Isolation Reagent (from serum; Thermo Fisher Scientific) overnight at 4°C and centrifuged (60 min, 10,000*g*; 70 min, 120,000*g*). Blood from humans and mice was collected into non-heparinized tubes (BD Vacutainer SST II Plus plastic serum tube; BD Biosciences), allowed to cloth for 30 min (room temperature) before serum retrieval by centrifugation (15 min, 1,000*g*). Samples were centrifuged (30 min, 2,000*g*) and EVs isolated with Total Exosome Isolation Reagent (from serum), according to the manufacturer. EVs were kept at –80°C until further analysis.

### EV isolation from human plasma samples by differential ultracentrifugation

Venous blood samples were collected into collection tubes with K$_2$EDTA (BD Vacutainer Plastic Blood Collection Tubes with K$_2$EDTA; BD Biosciences). Plasma was retrieved in the supernatant by centrifugation at 1,000*g* for 15 min at room temperature. Plasma samples were diluted in PBS (1:2), centrifuged at 2,000*g* for 30 min, followed by 45 min at 12,500*g*. Supernatants were ultracentrifuged for 2 h at 100,000*g*, after which pellets were resuspended in PBS, filtered (0.22 $\mu$m filter) and further ultracentrifuged at 100,000*g*, for 70 min. A last wash with PBS was performed, followed by ultracentrifugation at 100,000*g*, for 70 min.

### SEC

EVs were isolated from human plasma by SEC as previously described (de Menezes-Neto et al, 2015). Briefly, 10 ml of Sepharose CL-4B (GE Healthcare) were packed in a syringe and equilibrated with

PBS. 1 ml of human plasma, obtained from control or STEMI patients, was centrifuged at 2,000$g$ for 10 min, at 4°C, after which the supernatant was applied to the column. 20 fractions of 0.5 ml each were collected, using PBS as the elution buffer. Total protein was precipitated from each individual fraction with trichloroacetic acid (TCA), followed by WB analysis for CD63. CD63-positive fractions (fractions 6–8) were combined and WB analysis for Cx43 and CD63 was further performed.

### TEM

EVs were fixed with 2% PFA and deposited on Formvar-carbon coated grids (TAAB Laboratories Equipment). Samples were washed with PBS and fixed with 1% glutaraldehyde for 5 min. Grids were washed with water, contrasted with an uranyl-oxalate solution pH 7, for 5 min, and transferred to methyl-cellulose–uranyl acetate for 10 min on ice, as previously described (Soares et al, 2015; Martins-Marques et al, 2016). Images were collected using a Tecnai G2 Spirit BioTWIN electron microscope (FEI) at 80 kV.

### NTA

EVs were resuspended in 1 ml of PBS, after which NTA was performed using through a NanoSight NS300 instrument with a 488 nm laser and a sCMOS camera module (Malvern Panalytical), following general recommendations. Analysis settings were optimized and kept constant between samples and each video was analyzed to give the mean size and estimated concentration of particles. Data were processed using the NTA 3.1 analytical software.

### WB analysis

WB analysis was performed as previously described (Martins-Marques et al, 2020). For the analysis of CD63, CD81 or CD9, WB was performed in non-reducing conditions. Proteins of interest were visualized by chemiluminescence using a VersaDoc system (Bio-Rad). Densitometric quantification was performed in unsaturated images using ImageJ (National Institutes of Health). Antibodies against pS373-Cx43 were kindly provided by Dr Paul Lampe (Translational Research Program, Public Health Sciences Division, Fred Hutchinson Research Center). Antibodies against Cx43 (AB0016), CD63 (AB0047), GFP (AB0020), GAPDH (AB0049), and Calnexin (AB0041) were purchased from Sicgen. Antibodies against Flotillin-1 (sc-25506), AMSH (sc-98765), p-Tyr (PY99, sc-7020), p-Ser (16B4, sc-81514), p-Thr (H-2, sc-5267), pS368-Cx43 (sc-101660), CD81 (H-121, sc-9158), CD9 (C-4, sc-13118), and p62 (sc-28359) were obtained from Santa Cruz Biotechnology. Antibodies against Nedd4 (ab14592), p53 (ab131442), and Troponin T (ab33589) were from Abcam, whereas antibodies against p62 (5114S) and Alix (3A9, 2171S) were obtained from Cell Signaling. Antibodies against Ub (p4D1, 646302) were purchased from BioLegend, Tsg101 (4A10, GTX70255) was from Gentex, and Hsp90 (AC88, 386040) was from EMD Millipore. Antibodies against SART-1 (H9092-BO1P) were obtained from Abnova, and Hsc70 (ADI-SPA-815) were from Enzo Life Sciences. Antibody against USP8 (A302-929A) was obtained from Bethyl Laboratories, Inc. Antibody against Atg7 (A2856) was purchased from Sigma-Aldrich.

### IP

Cellular and EV lysates were prepared in radioimmunoprecipitation assay (RIPA) buffer, containing protease inhibitors (Martins-Marques et al, 2020). IP of Cx43 was performed in 500 $\mu$g total protein lysates by incubation with 0.5 $\mu$g goat anti-Cx43 (AB0016; Sicgen) overnight, at 4°C under agitation. Protein G Sepharose beads (GE Healthcare) were added and incubated for 1 h 30 min, at 4°C, followed by washing in RIPA buffer. Complexes were eluted in Laemmli buffer and denatured at 95°C, for 5 min.

### Enrichment of ubiquitinated proteins using glutathione S-transferase (GST) TUBEs

HEK293$^{Cx43+}$ cells were resuspended in RIPA buffer supplemented with protease and phosphatase inhibitors (protease inhibitor cocktail [Roche], 2 mM PMSF, 10 mM iodoacetamide, and 2 mM sodium orthovanadate). Lysates were briefly sonicated and centrifuged at 16,000$g$ for 10 min. Samples were incubated overnight at 4°C with 100 $\mu$g/ml of GST-TUBEs (UM101; LifeSensors), after which glutathione-Sepharose 4B beads were added, and incubations proceeded at 4°C for 2 h, followed by washing with lysis buffer. Pulled-down proteins were eluted in Laemmli buffer and denatured at 95°C, for 5 min before WB analysis.

### Immunofluorescence staining

Cells grown on fibronectin-coated coverslips were fixed in 4% PFA for 10 min and permeabilized with 0.2% Triton X-100 for 10 min. Specimens were blocked with 2.5% BSA for 20 min, followed by incubation with appropriate primary antibodies overnight, at 4°C. Alexa Fluor–conjugated secondary antibodies (Molecular Probes, Thermo Fisher Scientific) were incubated for 1 h, at room temperature. All solutions were made in 0.25% BSA. Nuclei were stained with DAPI. Specimens were mounted with MOWIOL 4-88 Reagent. For controls, primary antibodies were omitted. Images were analyzed by confocal microscopy in a Zeiss LSM 710 (Carl Zeiss AG).

### ELISA

Microplates were coated with anti-Cx43 capture antibody (#610062; BD Biosciences) overnight, washed and blocked with 1% BSA for 1 h. EVs were thawed on ice and SDS was added (1% final concentration). Samples were denatured (5 min, 95°C), sonicated and centrifuged (5 min, 1,200$g$). Supernatants were diluted to 0.1% SDS final concentration. Total protein was quantified (DC Protein Assay). EVs (5,000 ng/$\mu$l) and optical density standards (20–540 ng/ml of GST-tagged Cx43 soluble carboxyl-terminus) were added to the microplates for 2 h, washed before adding anti-Cx43 detection antibody (#710700; Thermo Fisher Scientific) for 2 h. After washing, anti-rabbit HRP-conjugated antibody was incubated for 20 min, washed and incubated 20 min with substrate solution (3,3′, 5,5-tetramethylbenzidine). Stop solution (1N HCl) was added and optical density determined at 450 nm using a microplate reader (background at 570 nm; Synergy HT, Biotek). EV-Cx43 levels were interpolated on a quadratic regression of the calibration curve

defined by standards' optical density, using GraphPad Prism 6 version 6.01 (GraphPad Software, Inc.).

## Decellularization of cultured cells

Cells were grown until post-confluency (10 d). Culture medium was removed, followed by incubation with decellularization buffer I (1 M NaCl, 5 mM EDTA, 10 mM Tris–HCl pH 7.4) for 1 h, at room temperature and buffer II (0.5% SDS, 25 mM EDTA, 10 mM Tris–HCl, pH 7.4) for 30 min, under agitation (Xing et al, 2015). After extensive PBS washing, ECM was scraped in PBS for TEM or RIPA buffer (150 mM NaCl, 50 mM Tris–HCl, 1% NP-40 and 0.1% SDS, pH 7.5) supplemented with protease inhibitors (protease inhibitor cocktail [Roche]), 2 mM PMSF, 10 mM iodoacetamide, and 2 mM sodium orthovanadate for total protein quantification (DC Protein Assay; Bio-Rad).

## Isolation of ECM-bound vesicles

Post-confluent monolayers were decellularized by incubation with 20 mM ammonium hydroxide/0.5% Triton X-100 for 5 min, at 37°C (Franco-Barraza et al, 2016). Samples were extensively washed with PBS and decellularized matrices were further incubated with enzyme buffer (80 U/ml collagenase Type II [Thermo Fisher Scientific], 10 U/ml DNase I [AppliChem], 1 mM CaCl$_2$, and 1 mM MgSO$_4$) for 1 h at 37°C. Digested ECM was centrifuged for 10 min at 300$g$, for 10 min, followed by 20 min at 16,500$g$ and 70 min at 120,000$g$. EV pellets were resuspended in PBS.

## Heart decellularization

Tissue was thawed, washed in PBS, cut into 2 × 2-mm pieces and incubated in hypotonic buffer (10 mM Tris HCl/0.1% EDTA, pH 7.8) for 18 h, at 25°C. Samples were washed in PBS (3×, 1 h), immersed in detergent solution (0.2% SDS/10 mM Tris HCl, pH 7.8) for 24 h. Tissues were washed in hypotonic buffer (10 mM Tris HCl, pH 7.8; 3×, 20 min) and incubated with DNAse (50 U/ml DNAse I/10 mM Tris HCl, pH 7.8) for 3 h, at 37°C. Decellularized hearts were washed in PBS (3×, 20 min), lysed in RIPA buffer supplemented with 0.9% SDS and sonicated (3×, 5″). Total protein was quantified (DC Protein Assay).

## EV-collagen binding in vitro

EVs purified from conditioned medium of HEK293[Cx43+] cells by differential ultracentrifugation (10 $\mu$g total protein) were incubated with 15 $\mu$g collagen S (Roche) for 1 h, at 37°C. Unbound EVs were washed in PBS by centrifugation at 16,500$g$ for 15 min. EV-collagen complexes were resuspended in PBS and further analyzed by WB or TEM, as appropriate.

## Sucrose permeability assay

Permeability of Cx43-containing EVs was assessed by a modified transport-specific fractionation assay, classically developed to evaluate permeability of liposomal Cx43 hemichannels (Bao et al, 2004). EVs (15 $\mu$g) were equilibrated in 0.4 M sucrose, layered over a discontinuous sucrose gradient (0.4–2.5 M) and ultracentrifuged (18 h, 160,000$g$). Sequential fractions were collected (1 ml, from top to bottom), followed by density assessment by Atago Uricon-N Hand-held Refractometer, and PBS washing by ultracentrifugation (70 min, 120,000$g$).

## Statistical analyses

Data represent are expressed as individual data points with mean ± SD. Independent variables were analyzed by Mann-Whitney test, whereas ANOVA (Tukey's post-hoc) or Kruskal–Wallis (Dunn's post-hoc) were used for multiple comparisons. Categorical variables were analyzed by chi-square test. All analyses were performed with GraphPad Prism 6.01.

# Data Availability

The data that support the findings of this study are available from the authors on reasonable request.

# Supplementary Information

# Acknowledgements

This work was supported by the European Regional Development Fund through the Operational Program for Competitiveness Factors (COMPETE) (under the projects PAC "NETDIAMOND" POCI-01-0145-FEDER-016385; HealthyAging2020 CENTRO-01-0145-FEDER-000012-N2323; POCI-01-0145-FEDER-007440, CENTRO-01-0145-FEDER-032179, CENTRO-01-0145-FEDER-032414, POCI-01-0145-FEDER-022122, FCTUID/NEU/04539/2013, UID/NEU/04539/2019, UIDB/04539/2020, and UIDP/04539/2020). This work was supported by the Project EVICARE (No. 725229) of the European Research Council to JPG Sluijter. T Martins-Marques was supported by PD/BD/106043/2015 and T Ribeiro-Rodrigues by PD/BD/52294/2013 from Fundação para a Ciência e a Tecnologia (FCT).

## Author Contributions

T Martins Marques: conceptualization, data curation, investigation, methodology, and writing—original draft, review, and editing.
T Ribeiro-Rodrigues: investigation.
SC de Jager: investigation and methodology.
M Zuzarte: investigation.
C Ferreira: investigation.
P Cruz: investigation.
L Reis: investigation.
R Baptista: investigation.
L Gonçalves: resources and methodology.
JPG Sluijter: conceptualization, resources, funding acquisition, and methodology.
H Girao: conceptualization, resources, data curation, formal analysis, supervision, funding acquisition, methodology, and writing—original draft, review, and editing.

**Conflict of Interest Statement**

The authors declare that they have no conflict of interest.

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
