## [Reviewer comments · Life Science Alliance]

Life Science Alliance

Myocardial infarction affects Cx43 content of extracellular vesicles secreted by cardiomyocytes

Tania Martins Marques, Teresa Ribeiro-Rodrigues, Saskia de Jager, Monica Zuzarte, Cátia Ferreira, Pedro Cruz, Liliana Reis, Rui Baptista, Lino Gonçalves, Joost Sluijter, and Henrique Girao

DOI: <https://doi.org/10.26508/lsa.202000821>

Corresponding author(s): Henrique Girao, University of Coimbra

Review Timeline:

Submission Date:	2020-06-19
Editorial Decision:	2020-08-07
Revision Received:	2020-10-09
Editorial Decision:	2020-10-12
Revision Received:	2020-10-12
Accepted:	2020-10-13

Scientific Editor: Shachi Bhatt

Transaction Report:

August 7, 2020

Re: Life Science Alliance manuscript #LSA-2020-00821-T

Dr. Henrique Girao
University of Coimbra
Center of Ophthalmology and Vision Sciences, Institute for Biomedical Imaging and Life Science (IBILI)
Faculty of Medicine
Coimbra 3000-548
Portugal

Dear Dr. Girao,

Thank you for submitting your manuscript entitled "Myocardial infarction affects Cx43 content of extracellular vesicles secreted by cardiomyocytes" to Life Science Alliance. The manuscript was assessed by expert reviewers, whose comments are appended to this letter.

The revised manuscript should address all of the Reviewers' concerns, including the concerns about Rev 2 with regards to the pattern of Cx43 in plasma derived EVs, resolve the concern about contaminants in kit-derived EV preps, and validation of the mechanisms of Cx43 sorting in the EVs in an adult cardiomyocyte cell culture.

In our view these revisions should typically be achievable in around 3 months. However, we are aware that many laboratories cannot function fully during the current COVID-19/SARS-CoV-2 pandemic and therefore encourage you to take the time necessary to revise the manuscript to the extent requested above. We will extend our 'scoping protection policy' to the full revision period required. If you do see another paper with related content published elsewhere, nonetheless contact me immediately so that we can discuss the best way to proceed.

Please note that papers are generally considered through only one revision cycle, so strong support from the referees on the revised version is needed for acceptance.

Thank you for this interesting contribution to Life Science Alliance. We are looking forward to receiving your revised manuscript.

Sincerely,

Reilly Lorenz
Editorial Office Life Science Alliance
Meyerhofstr. 1
69117 Heidelberg, Germany
t +49 6221 8891 414
e contact@life-science-alliance.org
www.life-science-alliance.org

B. MANUSCRIPT ORGANIZATION AND FORMATTING:

Reviewer #1 (Comments to the Authors (Required)):

This is an outstanding paper by experts in an emerging field that is shaping to be of major importance. The present report is a natural progression of the authors' work in this area. The work is conceptually innovative and the research design is very robust with appropriate controls and built in redundancies. The findings are highly compelling and potentially groundbreaking. The authors present high quality data that in my opinion convincingly demonstrate defective sorting of the classic gap junction protein Cx43 into cardiomyocyte-derived extracellular vesicles. They also rigorously highlight the importance of ubiquitin in the regulation of EV-Cx43 secretion and p62 in balancing act between Cx43 degradation and secretion. My only minor concern is that the paper, in general, needs significant editing. Specifically, the discussion section needs to be re-written with more concise focus on the novel findings of the paper.

Reviewer #2 (Comments to the Authors (Required)):

This study by Martins-Marques and colleagues investigate the mechanisms leading to Cx43 secretion in extracellular vesicles (EVs) secreted by cardiomyocytes following ischemia. They showed that ischemia affects Cx43 sorting into EVs. Additionally, authors studied molecular mechanisms involved in such sorting and show that ubiquitination promotes Cx43 sorting into EVs. Data presented in the manuscript are interesting. However some methodological points are critical. These points make the manuscript difficult to consider in the present version. Authors are invited to correct /comment some points in order to help reader and scientific community.

These points are listed here:

1-Tittle is misleading. As the International Society for EVs recommends, authors should state that EVs involved in this study are in fact small EVs so-called Exosomes. Indeed NTA data revealed that these Evs are small. Additionally authors do not observe the same pattern using the large EVs (ie 16500g pellet).

2-Analysis of circulating EVs have been performed on serum derived EVs. Such EVs are really different from plasma derived ones. Is Cx 43 still presenting the same pattern in plasma derived EVs?

Furthermore authors used a kit to "precipitate " EVs. It is now well recognized that such kits are precipitating also contaminants (protein aggregates). Authors should used state of the art methodology to obtain enriched fractions of so-called EVs ie small EVs by example Ultracentrifugation and or Size exclusion chromatography (SEC).

3-Molecular pathways analysis revealed that Cx43 sorting is independent of autophagy-mediated mechanisms using siRNA P62. Autophagy independent of p62 mechanisms exist and authors should prove using a complementary method (ie siBeclin-1 or TG5/7) that results are the same.

4-Authors state using WB that Cx43 is present in EVs. Authors should provide information using FACS analysis that CX43 is indeed into EVs and not at the membrane.

5- The detailed analysis of mechanisms leading to sorting of Cx43 in EVs is very well conducted and show for the first time how Cx43 is sorted into EVs. All these steps have been done in HEK cells. Authors should at least show that using adult cardiomyocyte cell culture or HL-1 cell line that mechanisms are conserved into such cell type.

Point-by-point answers

Reviewer #1 (Comments to the Authors (Required)):

This is an outstanding paper by experts in an emerging field that is shaping to be of major importance. The present report is a natural progression of the authors' work in this area. The work is conceptually innovative and the research design is very robust with appropriate controls and built in redundancies. The findings are highly compelling and potentially groundbreaking. The authors present high quality data that in my opinion convincingly demonstrate defective sorting of the classic gap junction protein Cx43 into cardiomyocyte-derived extracellular vesicles. They also rigorously highlight the importance of ubiquitin in the regulation of EV-Cx43 secretion and p62 in balancing act between Cx43 degradation and secretion. My only minor concern is that the paper, in general, needs significant editing. Specifically, the discussion section needs to be re-written with more concise focus on the novel findings of the paper.

We acknowledge the reviewer's encouraging and positive comments and thank the suggestions for improvement of the manuscript. In accordance, we have significantly edited the discussion section aiming at giving more focus and highlight the main novel findings of the present work.

Reviewer #2 (Comments to the Authors (Required)):

This study by Martins-Marques and colleagues investigate the mechanisms leading to Cx43 secretion in extracellular vesicles (EVs) secreted by cardiomyocytes following ischemia. They showed that ischemia affects Cx43 sorting into EVs. Additionally, authors studied molecular mechanisms involved in such sorting and show that ubiquitination promotes Cx43 sorting into EVs. Data presented in the manuscript are interesting. However some methodological points are critical. These points make the manuscript difficult to consider in the present version. Authors are invited to correct /comment some points in order to help reader and scientific community.

These points are listed here:

1-Tittle is misleading. As the International Society for EVs recommends, authors should state that EVs involved in this study are in fact small EVs so-called Exosomes. Indeed NTA data revealed that these Evs are small. Additionally authors do not observe the same pattern using the large EVs (ie 16500g pellet).

We acknowledge the pertinence of this issue. Despite the enormous effort of the scientific community devoted to EVs field to establish a consensual nomenclature, we recognize that this issue remains a major controversial topic, largely due to the rapid technical and scientific advances in this area, periodically challenging the established recommendations. Given that ISEV currently endorses the use of the generic term "EVs" (instead of "exosomes"), unless definitive prove of its unique endosomal origin is provided, we opt to maintain the use of the term "EVs" in the title and along the manuscript text, since we don't have clear evidence that the EVs we investigate stem from endosomes.

We do believe that the use of the term "exosomes" could be misleading to the readers, since in the present work, we did not establish specific markers of subcellular origin or biogenesis pathway. Taking into account the MISEV2018 guidelines, which recommend a nomenclature based on operational terms referring, for example, to the physical characteristics of EVs, in the beginning of our "Results section", we disclosed that small EVs are being analyzed (subsequently designated simply by EVs). Concerning the EVs released by cardiac endothelial cells we only evaluated the levels of Cx43 secreted into medium/large vesicles commonly named microvesicles (mainly because of the well-established importance of endothelial microparticles in cardiovascular pathophysiology). In this case, we observed that ischemia decreased the secretion of Cx43 into larger EVs derived from endothelial cells, but not in small EVs, advocating in favor of a cell type-specific response to ischemia.

2-Analysis of circulating EVs have been performed on serum derived EVs. Such EVs are really different from plasma derived ones. Is Cx 43 still presenting the same pattern in plasma derived EVs?

Furthermore authors used a kit to "precipitate " EVs. It is now well recognized that such kits are precipitating also contaminants (protein aggregates). Authors should used state of the art methodology to obtain enriched fractions of so-called EVs ie small EVs by example Ultracentrifugation and or Size exclusion chromatography (SEC).

We acknowledge the reviewer's concern and appreciate the suggestions. To address the issues raised by this reviewer, we performed additional experiments. For this, we isolated circulating vesicles from plasma samples obtained from control subjects and STEMI patients (1), using the provided approach to obtain serum EVs. Moreover, regarding the concern on the presence of non-EV contaminants, when the polymer-based kit was used to isolate circulating vesicles (Fig S8D), we also isolated EVs via both (2) SEC (Fig S8E) and (3) differential ultracentrifugation (Fig S8F) to purify plasma-derived EVs. In all cases (no 1-2-3), despite the low number of samples, we could confirm that the levels of Cx43 in circulating

EVs decrease in STEMI patients, when compared with control individuals (Fig S8D-F, page 11).

3-Molecular pathways analysis revealed that Cx43 sorting is independent of autophagy-mediated mechanisms using siRNA P62. Autophagy independent of p62 mechanisms exist and authors should prove using a complementary method (ie siBeclin-1 or TG5/7) that results are the same.

We do acknowledge Reviewer#2 for stressing this point. Although our aim was to demonstrate that ubiquitination of Cx43 can direct the protein either to EVs, for secretion, or autophagosomes, for p62-dependent degradation. The results presented in the manuscript support the hypothesis that ubiquitination during ischemia preferentially diverts Cx43 for lysosomal degradation through a mechanism that relies on p62 which recognizes ubiquitin moieties bound to Cx43. Following the reviewers' suggestion, we performed additional experiments in Atg7-depleted HL-1 cells, to block macro-autophagy, which we included in the revised version of the manuscript (page 7). Consistently with the data obtained in p62 knock-down cells, shRNA for Atg7 resulted in increased secretion of EV-Cx43 (Fig S3D). This further reinforces that, when degradation of Cx43 is compromised, the release of Cx43 into vesicles is promoted.

4-Authors state using WB that Cx43 is present in EVs. Authors should provide information using FACS analysis that CX43 is indeed into EVs and not at the membrane.

We acknowledge the pertinence of this issue, given the common use of FACS analysis to assess the presence of proteins in EVs, usually by using antibodies against proteins containing exposed domains. Cx43 is a membrane protein containing four transmembrane domains. Consistently, in a previous paper from our group (PMID: 26285688), using different complementary approaches, including surface biotinylation and trypsin-resistance assays, we demonstrated that Cx43 is embedded at the EVs membrane in the form of hexameric channels. Therefore, we believe that FACS analysis would not provide further relevant information regarding the localization of Cx43 at the EV membrane. Moreover, since all the antibodies to Cx43 are against the intracellular/ intravesicle domains of Cx43 this is technically challenging and not feasible for the small EVs.

5- The detailed analysis of mechanisms leading to sorting of Cx43 in EVs is very well conducted and show for the first time how Cx43 is sorted into EVs. All these steps have been done in HEK cells. Authors should at least show that using adult

cardiomyocyte cell culture or HL-1 cell line that mechanisms are conserved into such cell type.

We recognize the importance of cell type-specific mechanisms of EV secretion. However, the isolation and culture of adult cardiomyocytes is a challenging technique given the difficulty in isolating enough numbers of cells that provide levels of EVs suitable for further analysis. Due to the strong interaction between heart cells in the adult heart, and the firm binding of cardiomyocytes to the extracellular matrix network, it is difficult to effectively digest the tissue, without injuring the cells. Therefore, the yield of each cardiomyocytes isolation is reasonable for singular microscopy studies, that need lower number of cells, but not for experiments involving EV isolation, which will require high numbers of cultured EVs producing cells. Therefore, most of the experiments were performed in the 'easily-manipulated' cell line HEK293, but were replicated in the cardiac H9c2 (cardiomyoblasts) and HL-1 (atrial cardiomyocytes) cell lines: including Fig 2B and Fig S3A (in which we demonstrated the role of p62 in the secretion of HL-1 cells), Fig 4E-F, Fig S3G and Fig S5J-K (in which we show that ubiquitination regulates the sorting of Cx43 into EVs by the use of PMA and Heclin in HL-1 and H9c2, respectively), and in Fig S4F (in which we demonstrate the endosomal origin of EV-Cx43 in HL-1 cells overexpressing Rab5^{Q79L} mutants). Importantly, in the revised version of the manuscript, we included an additional experiment, performed in HL-1 cells, demonstrating that inhibition of autophagy, through the knockdown of Atg7, results in increased secretion of EV-Cx43 (Fig S3D).

October 12, 2020

RE: Life Science Alliance Manuscript #LSA-2020-00821-TR

Dr. Henrique Girao
University of Coimbra
Center of Ophthalmology and Vision Sciences, Institute for Biomedical Imaging and Life Science (IBILI)
Faculty of Medicine
Coimbra 3000-548
Portugal

Dear Dr. Girao,

Thank you for submitting your revised manuscript entitled "Myocardial infarction affects Cx43 content of extracellular vesicles secreted by cardiomyocytes". We would be happy to publish your paper in Life Science Alliance pending final revisions necessary to meet our formatting guidelines.

Along with the points listed below, please also attend to the following:

- please add the following statement to your conflict of interest section: The authors declare no conflict of interest.
- please use the [10 author names, et al.] format in your references (i.e. limit the author names to the first 10)
- please combine the supplemental material and methods section with the materials and methods section in the main manuscript, and combine the References into one section (supplemental + main manuscript)
- please rename the Methods as 'Materials and Methods'

A. FINAL FILES:

-- High-resolution figure, supplementary figure and video files uploaded as individual files: See our detailed guidelines for preparing your production-ready images, <https://www.life-science->

[alliance.org/authors](https://www.life-science-alliance.org/authors)

B. MANUSCRIPT ORGANIZATION AND FORMATTING:

Sincerely,

Shachi Bhatt, Ph.D.
Executive Editor
Life Science Alliance
<https://www.life-science-alliance.org/>
Tweet @SciBhatt @LSAJournal

Reviewer #2 (Comments to the Authors (Required)):

The paper by Martins-Marques and colleagues in the presented version (After revision) is significantly improved.

All concerns have been taking in account and the new version remain clear.

I have no further comments.

October 13, 2020

RE: Life Science Alliance Manuscript #LSA-2020-00821-TRR

Dr. Henrique Girao
University of Coimbra
Institute for Clinical and Biomedical Research (iCBR)
Faculty of Medicine
Coimbra 3000-548
Portugal

Dear Dr. Girao,

Thank you for submitting your Research Article entitled "Myocardial infarction affects Cx43 content of extracellular vesicles secreted by cardiomyocytes". It is a pleasure to let you know that your manuscript is now accepted for publication in Life Science Alliance. Congratulations on this interesting work.

DISTRIBUTION OF MATERIALS:

Again, congratulations on a very nice paper. I hope you found the review process to be constructive and are pleased with how the manuscript was handled editorially. We look forward to future exciting submissions from your lab.

Sincerely,

Shachi Bhatt, Ph.D.

Executive Editor

Life Science Alliance

<https://www.lsjournal.org/>
